artificial intelligence

academics, researchers, perceptions, opinions, integrative research concepts, interdisciplinarity

**Author for correspondence:**
Charlotte Teresa Weber
e-mail: charlotte.t.weber@uit.no

# Interdisciplinary optimism? Sentiment analysis of Twitter data

Charlotte Teresa Weber[1] and Shaheen Syed[2]

[1]Norwegian College of Fishery Science, UiT - The Arctic University of Norway, Tromsø, Norway
[2]Department of Information and Computing Sciences, Utrecht University, Utrecht, The Netherlands

 CTW, 0000-0003-4371-695X; SS, 0000-0001-5462-874X

Interdisciplinary research has faced many challenges, including institutional, cultural and practical ones, while it has also been reported as a 'career risk' and even 'career suicide' for researchers pursuing such an education and approach. Yet, the propagation of challenges and risks can easily lead to a feeling of anxiety and disempowerment in researchers, which we think is counterproductive to improving interdisciplinarity in practice. Therefore, in the search of 'bright spots', which are examples of cases in which people have had positive experiences with interdisciplinarity, this study assesses the perceptions of researchers on interdisciplinarity on the social media platform Twitter. The results of this study show researchers' many positive experiences and successes of interdisciplinarity, and, as such, document examples of bright spots. These bright spots can give reason for optimistic thinking, which can potentially have many benefits for researchers' well-being, creativity and innovation, and may also inspire and empower researchers to strive for and pursue interdisciplinarity in the future.

## 1. Introduction

Interdisciplinary research (IDR) involves activities that integrate more than one discipline with the aim to create new knowledge or solve a common problem. The interdisciplinary approach has gained popularity in science, education and policy over the last few years, and it is often advocated for solving today's complex problems and societal issues, such as climate change, biodiversity loss, food and water security, and public health issues [1,2]. It is hoped that IDR will help address these issues and create innovative solutions [3]. However, successfully crossing and integrating diverse fields and disciplines is not an easy endeavour.

IDR can face many challenges, from institutional [4] and cultural [5] to practical challenges [6,7]. Interdisciplinary work has also been reported to have lower funding success [8], can be challenging to

publish [9], and interdisciplinary journals are commonly perceived as less prestigious compared to single-disciplinary ones [10]. As a result, young scholars, following an interdisciplinary career-path, now fear to 'risk their careers', or even to 'commit career suicide' [11,12]. Researchers have also perceived their interdisciplinary experience as if they did not belong to a discipline, research community or research group. These researchers felt that they had to live without the comfort of expertise, while having to fight for identity, recognition and legitimacy within their work environment and among their peers [13]. This is, in part, because their background was too diverse or too broad to belong to a single discipline or to be considered an 'expert'. Many of these negative experiences and challenges have been, and continue to be, reported in the literature.

We believe that the continued propagation of challenges is counterproductive to improving IDR in practice. For example, negative wording as in 'less funding success', and 'career suicide' can easily create anxiety in (early-career) scientists and lead to a feeling of disempowerment, stopping them from even attempting IDR. While the study of such challenges and shortcomings is an important step when trying to improve IDR in the future, we argue that we also need to study the 'bright spots' [14] to harvest the full potential of IDR. These bright spots are examples in which people have had positive experiences with IDR and success stories of IDR, despite its challenges and barriers. We believe that the documentation of such bright spots and success can propagate optimism (understood here as the generalized expectancy that one will experience good outcomes [15]), which can further unlock creativity and innovation in interdisciplinary individuals and teams.

To study these bright spots, we applied machine learning algorithms for sentiment analysis to data collected from the social media platform Twitter. Previous research has shown that Twitter is generally used to broadcast thoughts and opinions [16]. Within academia, Twitter is also used to acquire and share real-time information, and develop connections with others. Furthermore, it has been shown that Twitter plays a significant role in the discovery of scholarly information and cross-disciplinary knowledge spreading [17]. Therefore, the aim of this study is to assess the perceptions of researchers and scientists on interdisciplinarity on a larger scale. As such, this study is in pursuit of bright spots within people's experiences shared on Twitter, with the ambition to create interdisciplinary optimism.

# 2. Material and methods

## 2.1. Defining the different modes of research

For the study of perceptions on interdisciplinarity, also transdisciplinarity and multidisciplinarity were considered. The terms *interdisciplinary*, *transdisciplinary* and *multidisciplinary* all describe different modes of research, which are conceptually visualized in figure 1 and understood and defined as follows [19]:

— **Interdisciplinary** research (IDR) refers to the integration of several unrelated academic disciplines that forces actors to cross boundaries with the goal to create integrated knowledge and theory.
— **Transdisciplinary** research involves the same integrative process as IDR, but includes non-academic participants.
— **Multidisciplinary** research involves multiple disciplines researching a common goal in parallel, but without integration or the crossing of subject boundaries.

Transdisciplinarity was included in this study because it involves an integrative process and is, therefore, similar to IDR. Multidisciplinarity was included because the term is sometimes used interchangeably with interdisciplinarity, while it is also often included in studies addressing interdisciplinarity [8,20,21]. In this study, the three concepts, interdisciplinary, transdisciplinary and multidisciplinary, will be referred to as *modes of research* and *integrative research approaches*.

## 2.2. Dataset

The dataset of publicly available tweets related to the three modes of research was constructed by using the Twitter search API. The Twitter search API returns tweets that match a specified search query. The returned tweets contain, for instance, the content of the tweet, the user who created the tweet, a description of that user and the unique ID associated with the tweet. To access the search API and query the tweets, the Python

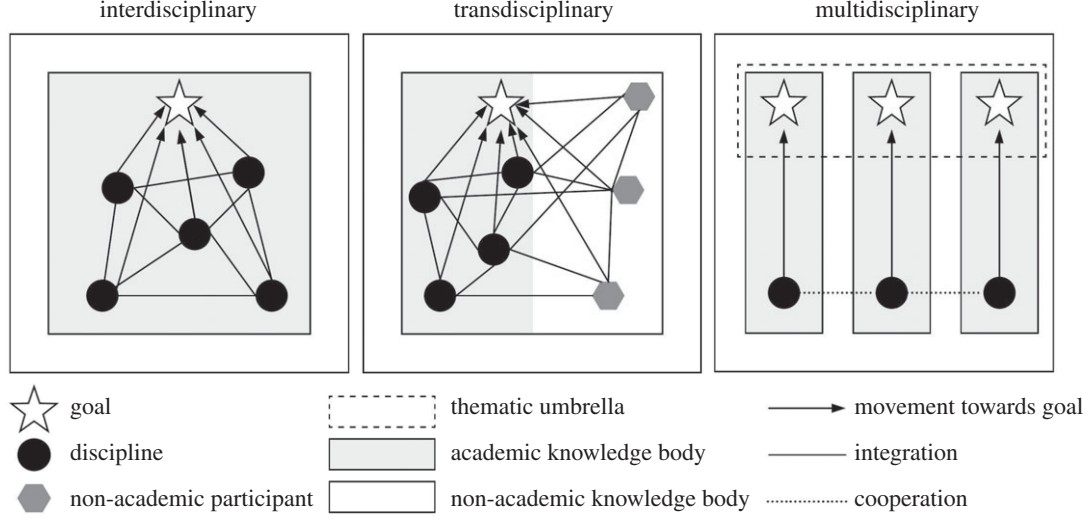

**Figure 1.** Overview of the three modes of research. Modified from [18].

**Table 1.** Overview of search queries used to retrieve tweets from the Twitter search API related to the three modes of research.

| mode of research | Twitter API search query |
| --- | --- |
| Interdisciplinary | 'Interdisciplinary OR #Interdisciplinary OR Interdisciplinarity OR #Interdisciplinarity' |
| Transdisciplinary | 'Transdisciplinary OR #Transdisciplinary OR Transdisciplinarity OR #Transdisciplinarity' |
| Multidisciplinary | 'Multidisciplinary OR #Multidisciplinary OR Multidisciplinarity OR #Multidisciplinarity' |

library *Tweepy*[1] was used. The used query strings to collect tweets related to the three modes of research are listed in table 1. The Twitter search API automatically returns all hyphenated variants of the search words, such as inter-disciplinarity and inter-disciplinary, eliminating the need to include such variations within the search queries. Since the Twitter search API only returns tweet data not older than 7 days, we collected tweets daily from week 32 (2017) up to week 33 (2018), spanning 53 weeks. During data collection, Twitter rolled out their expanded 280 character limit—previously 140 characters—which resulted in a dataset with 140 and 280 character limit tweets. It is important to note that the Twitter search API is not an exhaustive source of tweets, as not all tweets are indexed or available via the search interface. The full dataset of all collected tweets is made available in the electronic supplementary material, Data S1.

## 2.3. Audience of the dataset

Within this study, our aim is to make inferences regarding tweet sentiments associated with an academic or research domain. To identify tweets originating from a research domain setting, we filtered the dataset of all publicly available tweets for those from individuals who identify themselves as scientists (including all variations thereof, such as researchers or academics). To enable this, we used an adaptation of the systematic approach to identifying scientists on Twitter proposed by Ke *et al.* [22]. The filtering process essentially matches occupational classifications to the description field associated with the user account of the tweet. The description field is an optional field of a maximum of 160 characters, within which the user can describe herself; this is colloquially known as the user's bio.

We used the list of 322 scientific occupations (e.g. biologist, computer scientist and political anthropologist) compiled by Ke *et al.* [22]. This list was constructed by selecting the scientific occupations from: (i) the 2010 Standard Occupational Classification[2] (SOC) system released by the Bureau of Labor Statistics, United States Department of Labor; (ii) Wikipedia's list of scientific occupations[3] and (iii) the authors choice of adding generic occupations such as 'scientist' and 'researcher'.

---

[1]http://www.tweepy.org/.

[2]http://www.bls.gov/soc/.

[3]http://en.wikipedia.org/wiki/Scientist#By_field.

We augmented the list of 322 occupations by obtaining all the synsets (i.e. synonyms that share a common meaning) for each occupation from an online lexical reference system called WordNet [23]. Including the synsets, and excluding duplicate entries, resulted in a total list of 430 occupations related to a scientific or academic profession (see electronic supplementary material, Data S2). We then used regular expressions to match occupations with the user description field. This filtering approach identified, for example, tweets from a user describing himself as 'senior lecturer in human geography at the University of Liverpool' as valid for inclusion, and a description of a user describing herself as 'costume design and visual arts' as valid for exclusion. A random sample of 5000 user descriptions were manually examined to assess the inclusion and exclusion criteria, and adjustments were made to the regular expressions to enhance the filtering process (for instance, to capture American versus British spelling). We then excluded tweets that contained no user description text, since we were unable to identify if the user can be linked to an academic or research setting. On the one hand, the exclusion of tweets with no user description text might negatively affect the recall of relevant tweets. On the other hand, it positively affects the precision. In other words, we might not be able to include all the tweets from an academic or research setting (i.e. lower recall), but we can be more sure that the included tweets are all from the intended audience (i.e. higher precision).

## 2.4. Pre-processing tweets

Tweeting, the process of publishing a tweet, proceeds in the form of free text, often in combination with special characters, symbols, emoticons and emojis. This, in combination with a character limit, makes tweeters creative and concise in their writing, favouring brevity over readability to convey their message—even more so with the 140 characters limit. Thus, tweet data are highly idiosyncratic and several pre-processing steps were necessary to make the tweets suitable for sentiment analysis. These pre-processing steps included removing re-tweets and duplicated tweets, removing non-English tweets, normalizing user tags and URLs, handling contractions and repeated characters, the lemmatization of words, handling emoticons and emojis, removing numbers and punctuation, and creating appropriate bag-of-word features. All details concerning pre-processing are described in the electronic supplementary material, Text S1.

## 2.5. Creating the machine learning classifier

This paper employs a supervised machine learning approach to predict positive, neutral and negative sentiments from the tweets related to the three modes of research. Supervised machine learning essentially learns a sentiment classification model, called a classifier, from labelled tweet data, that is, tweets that have been labelled as positive, neutral and negative by human annotators. With the use of labelled data, the machine learning classifier learns that certain words convey, for example, positive sentiments when they more frequently occur in positively labelled tweets. The word *happy*, generally speaking, is used to convey a positive sentiment or feeling and tweets containing the word might be assigned a higher probability for the positive sentiment class. This is a somewhat basic and straightforward example but the classifier learns to assign every word—technically called a feature—a probability for each of the three sentiment classes. The tweet is thus a combination of features with corresponding probabilities and, ultimately, the classifier assigns the tweet a probability for the positive, neutral and negative class. The class with the highest probability is the inferred sentiment class. In essence, a supervised machine learning classifier is built or trained from labelled data and is applied to unlabelled data to predict or infer their label.

Several online repositories are available that contain human annotated tweet data. We combined several of such online repositories that serve as input data, called training data, to create or train the machine learning classifier. A total of seven different repositories were used, which contained a total of 71 239 labelled tweets, with 22 081 positive, 31 423 neutral and 17 735 negative tweets. Table 2 shows an overview of the datasets used to train the classifier, together with the frequency of tweets for the three sentiment classes, the domain or subject of the tweets, the number of human annotators used to label the tweets and a selection of research studies that have used the dataset. We provide descriptions of the seven datasets in the electronic supplementary material, Text S2. Note that the Twitter terms of service do not permit direct distribution of tweet content and so tweet IDs (references to the original tweets), with their respective sentiment labels, are often made available without the original tweet text and associated meta-data. As a result, we used the Twitter API to retrieve the full tweet content, the tweet text and the meta-data, by searching for the tweet ID.

**Table 2.** Overview of training datasets. For a full description of the datasets, see the electronic supplementary material, Text S2.

| dataset | positive | neutral | negative | total | domain | annotators | study |
|---|---|---|---|---|---|---|---|
| Sanders | 424 | 1996 | 475 | 2895 | Apple, Google, Microsoft, Twitter | 1 | [24–26] |
| OMD | 704 | — | 1192 | 1896 | #tweetdebate, #current, #debate08 | 3–7 | [27–29] |
| Stanford Test | 182 | 139 | 177 | 498 | consumer products, companies and people | 1 | [30–32] |
| HCR | 537 | 337 | 886 | 1760 | #hcr | 1 | [29] |
| SemEval-2016 | 3918 | 2736 | 1208 | 7889 | 100 different topics | 5 | [33] |
| SS | 1252 | 1952 | 861 | 4066 | major events | 1 | [34,35] |
| CLARIN-13 | 15 064 | 24 263 | 12 936 | 52 263 | 1% public available tweets | 1–9 | [36,37] |
| total | 22 081 | 31 423 | 17 735 | 71 239 | | | |

Some tweets appeared not to be available from the Twitter API and this, in some cases, resulted in the training datasets having fewer tweets than originally included in the published datasets.

## 2.6. Machine learning model selection

The labelled training datasets serve as input for building the machine learning classifier (i.e. training a model to classify tweets into positive, neutral and negative sentiments). Typically, with supervised machine learning, one would need sufficient data for each sentiment class to make good predictions on new (not used during training) tweets. We obtained a training dataset containing 71 239 labelled tweets, which can be considered a sufficiently large dataset for sentiment analysis.

Several (supervised) machine learning algorithms are suitable for the purpose of creating a sentiment classifier from labelled tweet data. Unfortunately, no consensus exists on what classification algorithm to use, since different studies have different datasets, perform different pre-processing steps, use different features, have incompatible performance measures or simply have different use cases. Thus, adopting one strategy that worked for a particular use case might not work for another. The current state of the art for sentiment analysis typically use algorithms based on neural networks [38,39]—also referred to as deep learning models—as can be seen from top-ranking teams during the SemEval 2017 competition [40]. The downsides of such winning entries are complexity, computational cost and the fact that they are highly tuned and optimized to achieve a high score on the task's performance measure. Besides neural network models, more traditional machine learning classifiers have also shown high accuracy and performance on sentiment classification tasks. They include Support Vector Machines (SVM) [41–43], logistic regression [24,43], Naive and Multinomial Naive Bayes [24,29,42,44] and Conditional Random Fields (CRF) [40]. Less complex neural networks, such as the Multi-layer Perceptron, have also been explored [44].

The aim of this paper is not to exhaustively explore the full suite of algorithms available but to use one that accurately predicts sentiments from tweets with reasonable complexity and computational time. Although complexity and computational time are hard to define concretely [45], we limit complexity to the basic machine learning and ensemble classification algorithms found in the Python library *Scikit-Learn* [46]. Additionally, we include a basic neural network, thus excluding very deep models and convolutional or sequence models. In terms of computational time, all selected models could be trained within a reasonable amount of time (10–30 min of wall clock time per model) on an Apple MacBook Pro with an i7 Processor and 16GB of internal memory. For example, the SVM with Gaussian kernel was not explored, because it was too time consuming to train a single model. A total of seven different supervised machine learning algorithms were considered: (1) SVM Linear Kernel; (2) Logistic Regression; (3) Multinomial Naive Bayes; (4) Bernoulli Naive Bayes; (5) Decision Trees; (6) ADA Boost and (7) the Multi-Layer Perceptron.

The dataset of 71 239 labelled training tweets was partitioned into two parts. The first part, called the training set, contained 80% randomly selected tweets used to train and validate the seven different algorithms. The second part, a random sample of 20% of the data called the test set, was used to test the

performance of the algorithms on tweet data that was not used during training. All seven different algorithms were applied to the training set with 10-fold cross-validation, which is a standard approach in machine learning [36]. During 10-fold cross-validation, the training set is partitioned into 10 parts, called folds, and training is done on nine folds with the remaining fold used to test the performance of the algorithm. This process is repeated 10 times, essentially creating 10 different classification models in which each model is tested against the remaining fold. Partitioning the data into folds is done on a stratified random basis, preserving the percentages of samples for each sentiment class. Additionally, we used a standard grid-search approach to establish the optimal performing parameter values for each of the seven algorithms. Since algorithms are parametrized and regularized by a set of parameters or hyper-parameters, finding the best performing values of these parameters can be obtained by trying out different combinations of values, called a grid-search. Other approaches, such as a random grid-search or Bayesian optimization [47] can also be considered, but were not employed in this study. The grid-search approach was combined with the 10-fold cross-validation method. For example, a grid search that tries out four different values for two separate parameters, combined with 10-fold cross-validation for a single algorithm results in $4 \times 4 \times 10 = 160$ different sentiment classification models. The different hyper-parameters and their explored parameter values are listed in the electronic supplementary material, table S1. The model that achieves the highest performance score (described in §2.8) is validated against the test set (i.e. remaining 20% of the data) to assess the performance of the model and its parameters on hold-out data; a way to measure the model's generalizability to unseen data.

## 2.7. Suitability training data

It is important that the training data from which we train the supervised machine learning classifier can appropriately infer sentiment classes for tweets containing the three modes of research (henceforth also called target tweets). By drawing on 71 239 training tweets (described in table 2), we captured a wide array of different sentiment expressions. However, specific sentiment expressions found within the target tweets can be absent from the training data, making accurate classification of the target tweets a challenging task. For instance, the word 'challenge' or 'difficult', in context, might not necessarily convey a negative sentiment within an academic setting. To mitigate this risk, we manually labelled a random subset of 1000 target tweets, stratified by the modes of research, as positive, negative or neutral. To have a common understanding of what positive, negative or neutral tweets constitute, we used the sentiment description text provided by the Amazon Mechanical Turk documentation for setting up a sentiment annotation project.[4] The Amazon Mechanical Turk is typically used as a crowd sourcing platform to annotate tweets for their sentiments [27,48,49]. Positive tweets embodied a happy, excited or satisfied emotion; negative tweets embodied an angry, upsetting or negative emotion; and neutral tweets did not embody much negative nor positive emotion. The 1000 labelled target tweets (containing specific sentiment expressions for interdisciplinarity, multidisciplinarity and transdisciplinarity) were added to training data to subsequently build the classification model. Additionally, to mitigate some risk of misclassification due to the absence of sample features in the training data, we used Laplace smoothing [50]—for the algorithms that allow smoothing. All pre-processing steps described in §2.4 were similarly applied on the training tweets and the target tweets so that the learned features of the training data could be correctly applied to the target tweets.

## 2.8. Calculating classification performance

Typically, for classification purposes, the performance of a model is assessed by the number of correctly predicted tweet sentiments in relation to incorrectly predicted tweet sentiments. We can discern three evaluation metrics: (i) precision; (ii) recall and (iii) F1. Precision measures how many of the tweets predicted to belong to a certain sentiment (e.g. positive) are actually positive. Precision, thus, measures how precise the predictions are. Recall measures how many of the positive tweets are captured by all of the predicted positive tweets, for example. Recall can be seen as a metric to evaluate whether the classification model can identify all the positive tweets from the complete dataset. There is a trade-off between optimizing recall and precision, and a summarized measure between the two is captured by the F1 score, a harmonic mean of precision and recall. We care equally about precision and recall and thus optimize both to achieve a high F1 score, which is an appropriate performance metric when having imbalanced classes (i.e. classes that are not completely

---

[4]https://docs.aws.amazon.com/AWSMechTurk/latest/RequesterUI/Create-Sentiment-Project.html.

**Table 3.** Classification performance metrics, precision, recall and F1 for the seven explored algorithms. Bold value indicates the highest F1 score on the test set.

| | training set | test set | | |
| --- | --- | --- | --- | --- |
| algorithm | F1 | precision | recall | F1 |
| SVM | 0.66 | 0.67 | 0.67 | **0.67** |
| Logisitic Regression | 0.66 | 0.66 | 0.66 | 0.66 |
| Multinomial NB | 0.64 | 0.65 | 0.65 | 0.65 |
| Bernoulli NB | 0.63 | 0.64 | 0.64 | 0.64 |
| Decision Trees | 0.55 | 0.56 | 0.56 | 0.56 |
| ADA Boost | 0.62 | 0.64 | 0.64 | 0.63 |
| ML-Perceptron | 0.59 | 0.60 | 0.60 | 0.60 |

represented equally). The trained model with the highest F1 score was ultimately used to predict tweet sentiments for the target tweets.

Additionally, after using the best performing classification algorithm on the target tweets, we created a random stratified dataset of 1000 target tweets (in addition to the 1000 manually labelled tweets reported in §2.7), stratified by the mode of research and now also by the inferred sentiment label. This dataset was manually labelled for the true sentiment class, and the performance of the classification algorithm was measured against it. In doing so, the classification performance on the target tweets, besides the training tweets, could be measured. We report the precision, recall and F1 measures as previously described.

## 2.9. Inspecting tweets

To get a better understanding of the content and context of researchers' positive, neutral or negative sentiments regarding interdisciplinary, transdisciplinary and multidisciplinary, a random set of tweets (in the range of $n = 2000$ or less, depending on the number of tweets classified into a specific sentiment) for each sentiment class and for each mode of research was manually examined. During the examination, one of the authors read the tweets and noted down what researchers felt positive, neutral or negative about (e.g. a conference, a publication, collaboration, etc.) and why (e.g. because of inspiring talks, creative outcomes, rewards, etc.). The reasons (what and why) for the expressed sentiment were collected and summarized into common themes, as presented in the results.

# 3. Results

## 3.1. Classifier performance

Table 3 shows the three evaluation metrics (i.e. precision, recall and F1) for the seven explored classification algorithms that were applied to the training (80%) and test (20%) partitions of the labelled tweet data. The obtained F1 score on the training set and test are very similar, an indication that none of the models are overfitting. A model that overfits the training data generally performs worse on the test data as it is unable to generalize new, unseen data. The results within the test set columns, and specifically, the reported F1 metric show that the created SVM classifier performs best. The hyper-parameters for the seven explored algorithms are described in the electronic supplementary material, table S2.

As a result, we used the trained SVM classifier to infer a positive, neutral or negative sentiment from the target tweets, that is, the collected tweets related to interdisciplinary, transdisciplinary and multidisciplinary. Evaluating the trained SVM classifier on a manually labelled subset of the target tweets (§2.8) resulted in an precision score of 0.84, recall score of 0.83 and F1 of 0.83.

## 3.2. Sentiment classification

The largest set of data, over 47 000 tweets, was collected for the interdisciplinary mode of research, followed by multidisciplinary and transdisciplinary with the least tweets (figure 2). Neutral sentiment was most common in all three modes, including more than 50% of the interdisciplinary tweets, and around 80% of the

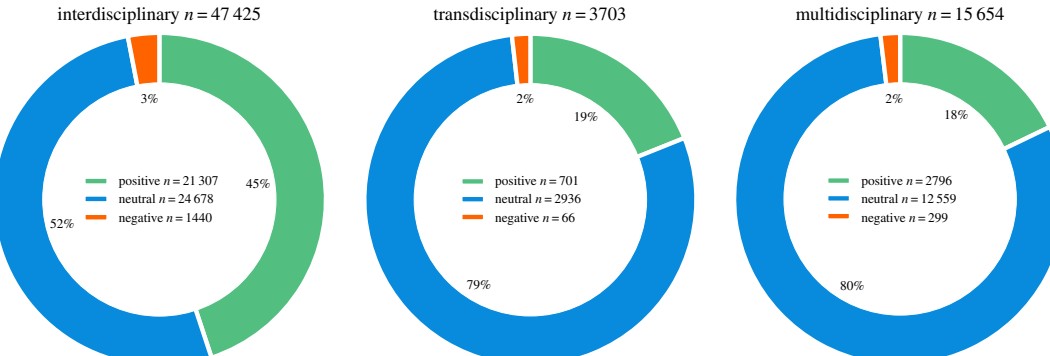

**Figure 2.** Frequency of tweets by sentiment for the three modes of research.

**Table 4.** Number of unique user names for each mode of research and sentiment class and the ratio for the number of tweets per unique user.

| mode of research | sentiment | unique users | ratio |
|---|---|---|---|
| interdisciplinary | negative | 1259 | 1.13 |
| | neutral | 15 466 | 1.58 |
| | positive | 15 022 | 1.41 |
| | total | 27 265 | 1.72 |
| transdisciplinary | negative | 56 | 1.18 |
| | neutral | 1967 | 1.49 |
| | positive | 584 | 1.20 |
| | total | 2380 | 1.56 |
| multidisciplinary | negative | 275 | 1.08 |
| | neutral | 8741 | 1.43 |
| | positive | 2445 | 1.14 |
| | total | 10 752 | 1.45 |
| all three modes of research | total | 37 213 | 1.78 |

transdisciplinary and multidisciplinary tweets, whereas, all three modes of research contained more positive than negative tweets. Interdisciplinary tweets contained the largest percentage of positive tweets with almost half of the tweets being positive (45%), while the transdisciplinary and multidisciplinary modes of research contained less than 20% positive tweets. The percentage of negative tweets was relatively low, with similar percentages between 2% and 3% among all three modes of research (figure 2).

The number of tweets per week showed no drastic increases or decreases over the study period, with two exceptions: A small drop in the number of tweets occurred around week 52 in 2017, during what is typically a holiday period (see the electronic supplementary material, figure S1). The number of tweets also dropped at around week 23 in 2018 because tweets were not collected for three days due to server downtime. The proportion of positive, neutral and negative tweets stayed relatively stable over the study period (electronic supplementary material, figure S1).

## 3.3. Number of users and tweet

The tweets from all three modes of research originated from over 37 000 different unique users, with a tweet ratio of 1.78, i.e. the number of tweets per unique user (table 4). Interdisciplinary tweets showed the highest total tweet ratio, while multidisciplinary tweets showed the lowest total tweet ratio. Within all three modes of research, neutral tweets showed the highest tweet ratio, and positive tweets had a higher tweet ratio then negative tweets (table 4).

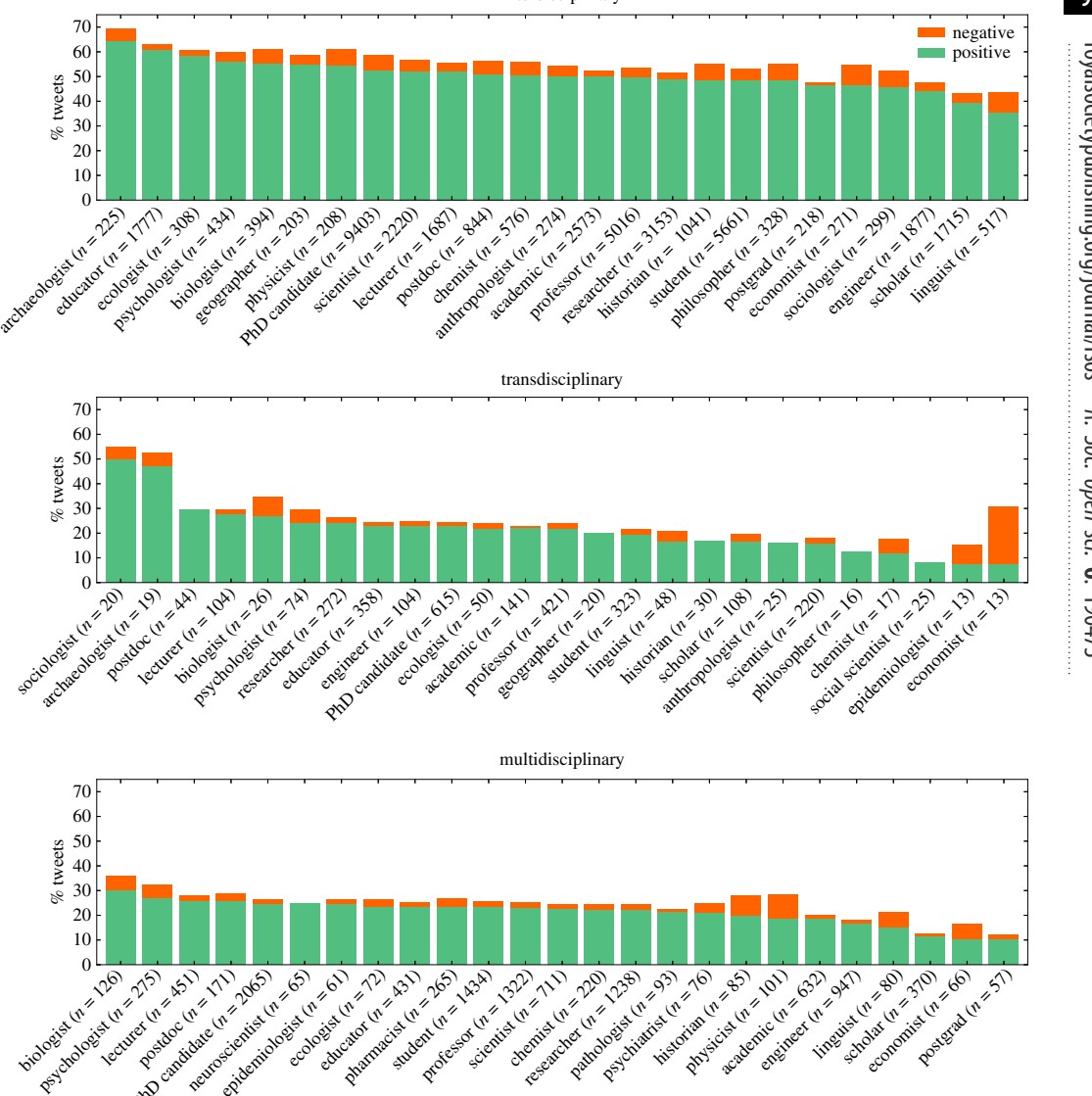

**Figure 3.** Top-25 occupations with the highest percentage of positive tweets for the three modes of research; number of tweets between parentheses. Note that mappings to occupations, as described in §2.3 are not mutually exclusive.

## 3.4. Users' occupation

Figure 3 shows the top-25 occupations with the highest percentage of positive tweets for the tweets related to three modes of research. All three modes of research contain generic, field-unspecific occupations, such as researcher, scientist, lecturer, scholar, academic and educator among the top-25 occupations. Other academic occupations that contributed with a high number of positive tweets are professor, student, PhD candidate, postdoc and postgrad. The field- and discipline-specific occupations in the top-25 cover a range of disciplines from the natural sciences (e.g. biologist, physicist, neuroscientist), social sciences (e.g. sociologist, economist) and humanities (e.g. archaeologist, anthropologist, linguist). Many of the same occupations are present among all three modes of research, such as engineer, biologist, chemist and economist. The top-25 occupations with the highest percentage of positive tweets for interdisciplinary and transdisciplinary include more occupations from the social sciences and humanities, whereas multidisciplinary tweets contain more natural science occupations.

## 3.5. Frequency of user tags, URLs and emojis in the tweets

User tags (@) were used similarly, with around one tag per tweet, within all three modes of research and for all three sentiment classes. Only positive multidisciplinary tweets showed slightly more user tags

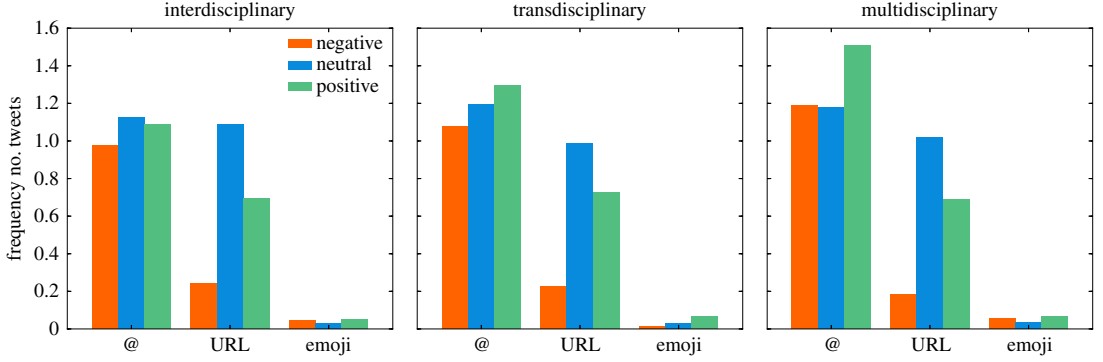

**Figure 4.** Relative counts of user tags (@), URLs and emojis within the tweets of the three modes of research.

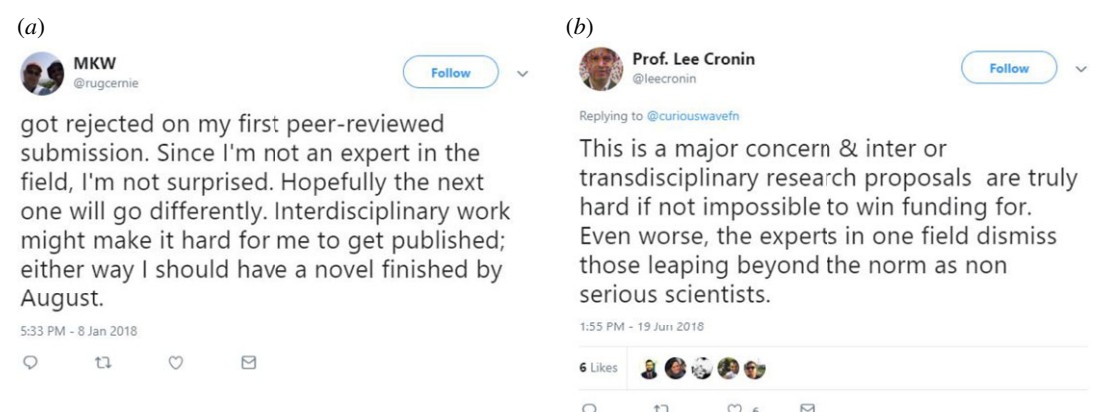

**Figure 5.** Examples of negative tweets. (*a*) Manuscript rejection and (*b*) funding challenges.

compared to the other sentiments and modes (figure 4). URLs were used most frequently in neutral tweets for all three modes of research, with an average of one URL per tweet, highlighting their informative character, while negative tweets showed the lowest use of URLs in all sentiment classes within all three modes of research. Emoji usage was relatively low compared with the frequency of user tags and URLs in all three modes of research, with around one emoji in every 10th tweet. Generally, emojis were used more frequently in positive and negative tweets, than in neutral ones (figure 4).

## 3.6. Tweet content

### 3.6.1. Negative tweets

Several of the negative tweets within all three modes of research contained explicit language. In the negative tweets, researchers explicitly state that they do not enjoy an integrative approach. Additionally, tweets reflect the challenges that are associated with integrative research in practice and being an integrative scholar. Additionally, negativity is expressed by criticizing the people or the system that discourage integrated research approaches.

*Negative interdisciplinary* tweets most often discuss challenges of IDR, such as rejections by peer-review (figure 5*a*), a lack of integration, communication problems across disciplines and difficulties in securing funding. Criticism of the existing institutional system of disciplinary departments was also mentioned. The need for more IDR was repeatedly mentioned, while the lack of acknowledgement and respect for IDR was also a recurring topic. There were only very few *negative transdisciplinary* tweets (*n* = 66). However, the tweets related mostly to challenges within transdisciplinary work, being a transdisciplinary scholar, and funding concerns (figure 5*b*). *Negative multidisciplinary* tweets often related to the challenges and problems in healthcare, patient care and treatment. Also education and teaching were a recurring theme, in which the lack of multidisciplinary perspectives and teaching approaches was criticized. Challenges in publishing were also mentioned.

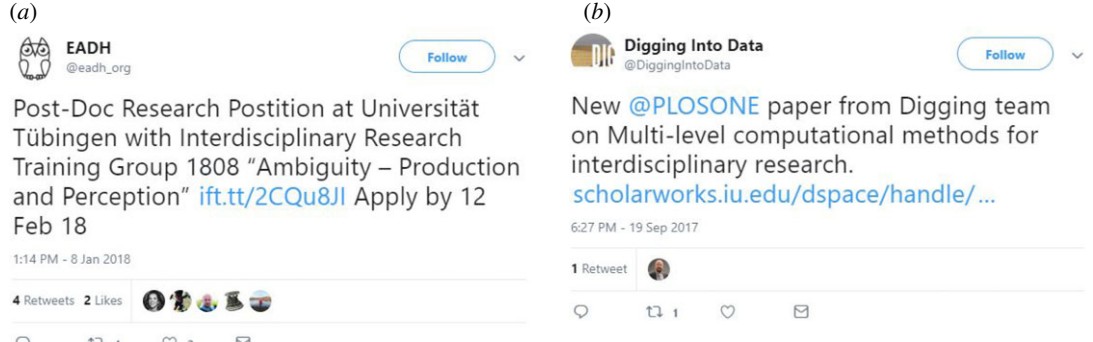

**Figure 6.** Examples of neutral tweets. (*a*) Job announcement and (*b*) paper publication.

### 3.6.2. Neutral tweets

The neutral tweets were mostly informative and many of them shared and advertised publications, websites or job announcements. *Neutral interdisciplinary* tweets frequently shared information referring to job postings, PhD positions, open calls for applications, news, blog posts, pod-casts, discussions or researchers announcing a paper publication (figure 6*a*,*b*). A similar pattern was visible in the *neutral transdisciplinary* tweets, with many tweets referring to informative topics such as job posting, articles and news publications. Similar results were also found within the *neutral multidisciplinarity* tweets, which mainly referred to websites, news, events, articles, paper publications and job postings.

### 3.6.3. Positive tweets

The content of positive tweets within all three modes of research showed enthusiasm and excitement by complimenting and praising different studies, lectures, approaches and discussions. Many of the *positive interdisciplinary* tweets were positive about conferences, seminars, symposia and workshops, in which researchers felt excited and thought that these events were interesting and useful. In particular, researchers were positive about meeting like-minded scholars and listening to inspiring talks, discussions and thoughts during conferences, seminars and workshops. Researchers described their participation in such events as inspiring and exciting, and they felt lucky to have participated. In many of the tweets, researchers also described having fun learning, enjoying listening to others and appreciating critical interdisciplinary discussions. In many tweets, researchers were also grateful to the organizers and participants of these events.

The tweets also expressed positivity towards research communities, teams and collaboration (figure 7*a*). Researchers were happy and excited about effective, successful and inspiring team work, collaboration and cooperation in integrative research projects and studies. Collaboration was also enjoyed during paper writing, seminars and workshops. Many expressed appreciation for cooperation and collaboration, and described how interdisciplinary collaboration can add value to advance research and understanding. The term 'bridge building between disciplines' was often mentioned as a goal to strive for. Many of the positive tweets reported overall positive experiences, feelings or praised interdisciplinary work and results (figure 7*b*,*d*).

Researchers expressed their appreciation of the value and importance of integrative work, and were often in support of integrative research, as from their perspective, it can provide important solutions and promising results in different fields. For example, the fields of neuroscience, cancer research, political science, fisheries science, engineering, gender studies, cognitive science, computer science and archaeology were mentioned. Researchers also expressed their explicit support for a certain interdisciplinary approach, conveyed their excitement about a study, and highlighted how they believe that a certain approach can address a particular challenge or solve a certain complex problem. Many described interdisciplinary work as impactful, excellent, creative and innovative, with potential for new discoveries. Others highlighted the strengths of integrative approaches, and how integrative research could potentially benefit sustainability and human well-being, such as patient care and mental health.

Tweets about the successful acquisition of funding and research grants for IDR by individuals, teams and research groups were also shared several times (figure 7*c*). Some tweets showed excitement about having an interdisciplinary job, getting a new job within an interdisciplinary field or the successful

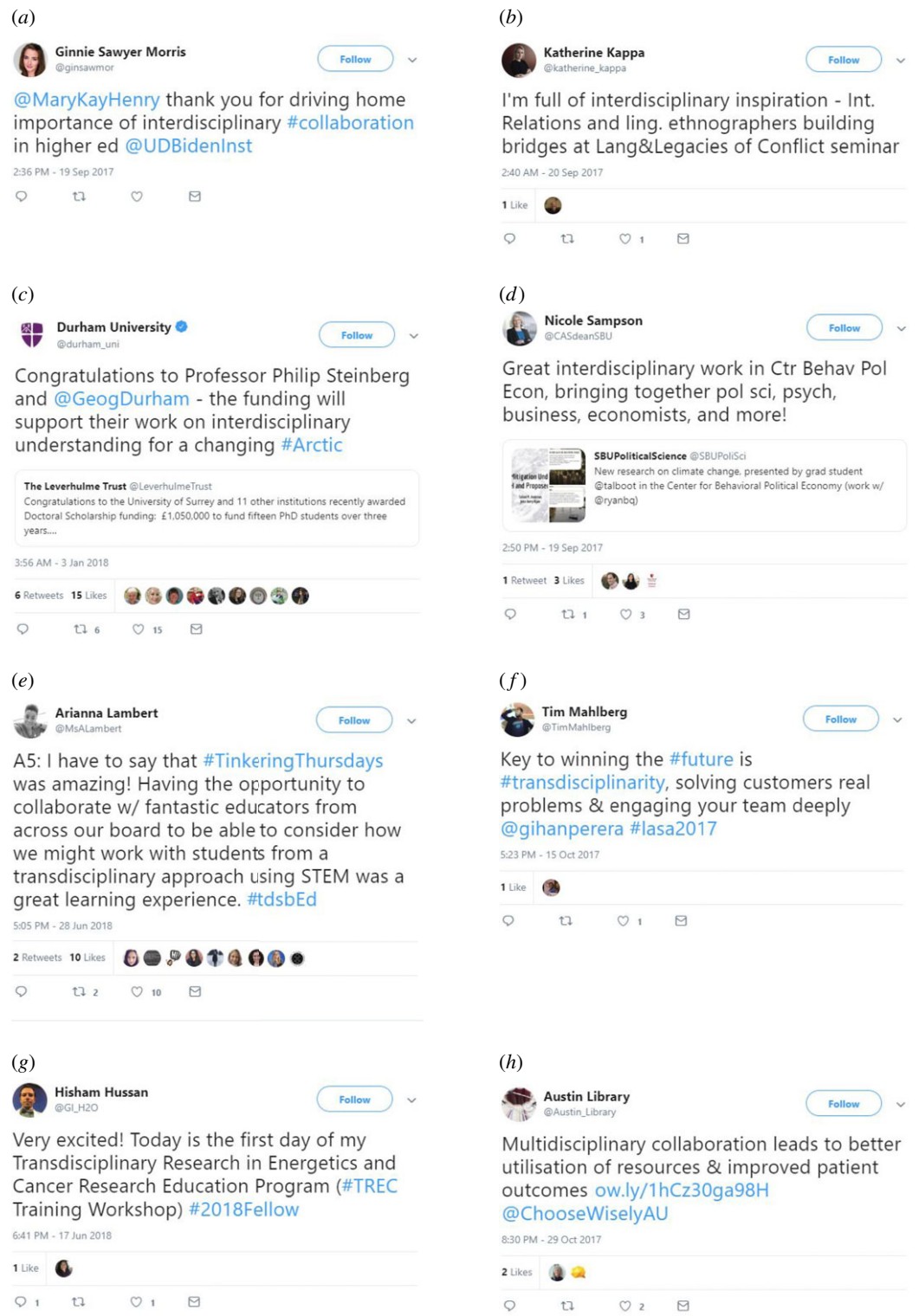

**Figure 7.** Examples of positive tweets referring to all three modes of research (interdisciplinary (*a–d*), transdisciplinary (*e–g*) and multidisciplinary (*h*)). (*a*) Importance of collaboration, (*b*) inspiring seminar, (*c*) funding acquisition, (*d*) work appraisal, (*e*) collaboration and education, (*f*) appreciation, (*g*) excitement, (*h*) value of collaboration.

completion of an interdisciplinary PhD. Many described their work and research as fun and rewarding, and felt proud of their achievements and accomplishments. Researchers also appraised interdisciplinary universities and the value and importance of interdisciplinary education and training, and the benefits thereof. For example, researchers described the benefits of newly learned abilities and skills through interdisciplinary work, interdisciplinary training that functioned as an eye-opener towards other fields,

and how an interdisciplinary perspective can provide additional food for thought. Interdisciplinary training was also described as a way for researchers to open up alternative career paths.

Researchers also commonly shared their excitement and happiness about the successful publication of an interdisciplinary paper, book, book chapter, news article or blog post about IDR or interdisciplinary experiences. Some were also joyful about sharing news over the recognition of their IDR and teaching through awards and prizes.

*Positive transdisciplinary* tweets expressed positive sentiment about very similar topics found in interdisciplinary tweets, such as publication and funding successes, positive experiences from conferences and discussion, and the value of transdisciplinary teaching and education (figure 7e). Also, positive experiences from the involvement in transdisciplinary research projects were shared, highlighting the value and importance of the work (figure 7f,g).

Many of the *positive multidisciplinary* tweets referred to healthcare and medical topics (figure 7h). Other topics included, similarly to inter- and trans-disciplinary tweets, good experiences from teams and teamwork, positive conference experiences, a general positive attitude and excitement towards multidisciplinary work and success stories, such as winning an award.

*Key successes and positive experiences.* In summary, the positive tweets demonstrated a number of key successes and documented positive experiences for the three modes of research.

Key successes included:

— attaining advanced skills;
— successful publications (papers, books, etc.);
— acquisition of funding and research grants;
— production of creative, innovative and impactful research;
— improved research practice and research results through effective and successful teamwork, collaboration and cooperation;
— research results provide useful solutions and can be of value to science and society; and
— recognition of scientific work through awards and prizes.

Descriptions of positive inter-, trans- and multi-disciplinary experiences can be summarized as follows:

— Conferences, seminars and workshops provide valuable insights and inspiring talks.
— Meetings with other inter-, trans- and multi-disciplinary scholars provide thought-provoking and inspiring discussions.
— Work and research is fun, exciting and rewarding.
— Training and learning opportunities are interesting and valuable.

# 4. Discussion

We used sentiment analysis to explore the opinions of researchers towards different modes of research (including inter-, trans- and multi-disciplinarity) communicated via the social media platform Twitter. Twitter provides an immense resource of text-based data covering a very large group of people [51], with an average of 328 million active monthly users who broadcast their thoughts and opinion as tweets [16]. As such, Twitter offers an abundance of easily accessible data, which has resulted in the rise and development of machine learning techniques for sentiment analysis of tweets [52]. To date, sentiment analysis on Twitter data has been conducted across a variety of disciplines and topics, ranging from computer science to environmental and medical sciences [53–57]. In addition, Twitter data have also been used to save lives during earthquakes, while organizations such as the United Nations collaborate with Twitter to achieve sustainable development goals,[5] for example, to monitor outbreaks of diseases [58]. Hence, it is increasingly being recognized that tweets can provide valuable information and insights into people's lives, health and opinions.

In our study, over 70 000 tweets were collected for the different modes of research over the time span of 53 weeks. Over 37 000 scientists and researchers were captured in our dataset (table 4), which demonstrated that there is a large scientific community that is interested and actively engaged in the discourse of integrative research concepts. Proportionally, interdisciplinarity appears to be the most popular research concept with the largest number of collected tweets, which could be related to the general IDR 'break out' over the last couple of years [59].

---

[5]https://developer.twitter.com/en/developer-terms/more-on-restricted-use-cases.

*Negative tweets.* The negative tweets highlighted the challenges of integrative research that people have experienced. It is often the integrative nature of these approaches that gives rise to challenges for the researchers involved, because disciplinary boundaries have to be crossed, which can introduce institutional and cultural issues [4,5]. The detection of negative opinions within the inter- and trans-disciplinary tweets was therefore foreseen. By contrast, multidisciplinary tweets, reflecting a non-integrative research concept, were expected to have fewer negative tweets because multidisciplinarity is often perceived as being 'easier'. Interestingly, this hypothesis could not be confirmed in this study, nor in a similar study by Bruun *et al.* [60].

Only very few of the tweets were classified as negative (2–3%), which stands in contrast to the larger literature, in which the challenges and difficulties of integrative research are often propagated and discussed [4,5,7,8]. Therefore, this study highlights how a sentiment analysis can offer new insights into researchers' opinions, and has the ability to identify perspectives on a larger scale that contrast the common (negative) perception and experiences typically found in the literature. However, the articles and publications in the literature are usually published by selected individuals and do not necessarily reflect the opinion of the wider research community, whereas our study captured opinions of thousands of different individuals (table 4). In addition, not all researchers may want to share their negative experiences with their friends and colleagues on Twitter, as it is often easier to share and disseminate one's successes rather than one's failures, which could further explain the low number of negative tweets in our dataset. Yet, from a scientific perspective, it may be perceived as more valuable to identify and analyse challenges within research practice, with the aim to understand and overcome these challenges. However, human brains tend to have a tendency towards negative bias [61], which means that people usually have higher sensitivity to negative information. Despite the description of positive examples and experiences in the literature, e.g. [62,63], the negative ones might be more likely to be remembered by researchers, and could, therefore, potentially have a counterproductive effect.

*Neutral tweets.* The majority of tweets found in this study were neutral and mostly informative, which did not reflect any particular perception or sentiment. In these cases, Twitter was being accessed as a dissemination and advertising platform, rather than a way to express an opinion, because it is generally a cheap and easy way to spread information. This is not surprising, considering that Twitter is increasingly used as a source of real-time information from news channels, politics, business, science and entertainment, and its personal use by individuals and organizations lies at the core of Twitter's utility to express thoughts, share information and connect with friends [64].

Typically, scholars, who are the population of this study, tweet neutral information, resources and media [64], and this explains the high amount of neutral tweets with URLs found in this study (figures 2 and 4). Twitter is also increasingly used as a teaching and communication platform by instructors [65], who have been found to have a higher credibility among students when their tweets are professional [66]. Such educational tweets most often express no sentiment and are of a neutral nature. There has also been a gradual shift within the scientific community towards increased communication and dissemination of research and scientific results through social media. A possible explanation could be the increasing demands for dissemination by funding agencies such as the European Commission [67]. For researchers, the Internet has become a useful way to disseminate and promote events and publications and Twitter offers a quick and easy option to disseminate scientific results, contribute to a discussion and increase visibility via hashtags [64]. The use of Twitter by researchers for these purposes is also apparent in our results.

*Positive tweets.* Positive tweets demonstrated experiences of success, such as successful funding acquisitions, interdisciplinary publications, successful and positive teamwork experiences, and successful interdisciplinary projects (figure 7). These aspects of interdisciplinarity (e.g. funding, publications, teamwork) are often perceived as rather challenging [6–9]. Thereby, our results are demonstrating real-life examples of how the positive opposite of what is commonly feared is possible and attainable within IDR.

A large number of the tweets were positive, within which users expressed positive experiences and perceptions. The high frequency of user tags (@) (figure 4) implied discussions between people, projects and institutions, e.g. within a circle of friends, between co-workers or in connection with a shared field of research or project. Participants from integrative projects were often found to have positive experiences based on teamwork and collaboration with other participants [18,60], which is also indicated in our results (figure 7*a*). The positive tweets were likely to have originated from people who were actively involved in interdisciplinary projects themselves (figure 7*g*), which is also the group of researchers that has been found to describe their work as positive most of the time [60].

We hypothesize that many of the younger generations of scientists within our dataset perceive interdisciplinarity as mainly positive and beneficial, more intellectually interesting, and more practically important. This is due to the fact that we found a high number of positive tweets from students, PhD candidates, postdocs and postgrads (figure 3). This is in line with the literature, where younger researchers show higher rates of interdisciplinarity compared to tenure track researchers and professors [12]. These scholars are likely to share their enthusiasm and success through a positive attitude and discourse on integrative research, especially women [68]. The perceptions of early career researchers could be dominating the positive discourse on Twitter since 24–35 year olds make up the largest age group of Twitter users [69]. On the contrary, we also found a high number of positive tweets from professors (figure 3), indicating that also senior researchers present on Twitter are in support of interdisciplinarity. However, age and gender distributions were not investigated in this study and cannot be confirmed at this stage.

The occupations in our study that feel most positive about interdisciplinarity include biologists, epidemiologists and chemists. Disciplines such as biology and chemistry have also been found to be among the more interdisciplinary ones compared to other disciplines [70]. However, we also found occupations more related to the social sciences and humanities (e.g. sociologists, anthropologists and archaeologists) that feel very positive about integrative research approaches, even though social science disciplines were found to be less interdisciplinary [70]. Hence, positive perceptions of interdisciplinarity may be more dependent on the individual, rather than on the interdisciplinary efforts within a discipline.

*Reason for optimism?* Overall, our study revealed that researchers have mostly positive perceptions about multi-, inter- and trans-disciplinarity (figure 2). This study also demonstrates many examples of positive experiences that were created through successful funding, accepted publications, interesting research outcomes, and effective teamwork and collaboration, besides other aspects. This highlights that there are, indeed, many 'good experiences' and 'bright spots' to be found within these research practices. It also shows the value of this Twitter analysis, as some of these experiences may not be shared to the same extent within the literature as such. For example, publications seldom cover success stories regarding the acquisition of funding for an interdisciplinary project or the experiences of conference participants. Hence, this Twitter sentiment analysis is able to capture and quantify interdisciplinary experiences from a different perspective.

We believe that the findings of this study demonstrate and document positive experiences and opinions, and as such, give reason for optimism within integrative research approaches. Hence, this study is a first step towards building interdisciplinary optimism. The continued documentation and propagation of such 'bright spots' and successes could further increase optimistic thinking about integrative research, which may have many potential benefits.

Optimism can increase people's psychological and physical well-being [15], and facilitate and increase creativity in individuals and teams [71]. Creativity is also closely linked and thought to play an important role in people's innovation capacity [72,73]. This makes it a key aspect for integrative research approaches, which aim to show high innovation potential. Positive thinking and optimism are also beneficial to team work—a crucial part of most integrative approaches—and can have positive effects on team-level cooperation, collaboration and overall team outcomes [74]. Hence, the findings of this study could potentially contribute to the future success of integrative research through their propagation of optimism. Therefore, we believe that it is important to make these results visible to the wider research community through publication and dissemination, and that additional positive experiences, as well as studies of bright spots, should be shared and propagated. Thereby, interdisciplinary researchers are encouraged to follow this example and participate in interdisciplinary discourse and the study of bright spots to support integrative research practices from an optimistic perspective in the future.

# 5. Limitations and future work

An inherent limitation of Twitter is that it is not representative of the whole population and our study could be expanded to compare and contrast our results with other communication media in the future. In addition, we provided only a snapshot in time, and therefore, we would encourage the study of the long-term trends of public sentiment towards integrative research approaches, along with additional investigations into the distribution of age and gender among Twitter users. We included tweets exclusively in English because it is the most common language found on Twitter, but this, potentially, excludes all non-English discourse on interdisciplinarity.

The detection of sarcasm and irony remains a difficult and challenging task, and is a limitation of this study. However, this limitation is somewhat mitigated by drawing on a large dataset of over 70 000 tweets. In addition, we assumed people to be truthful in their tweets. There is also a risk that the use of the words multidisciplinary, interdisciplinary and transdisciplinary may not be based on a solid understanding of these terms. It is also extremely difficult to assess whether a person has a correct understanding of the terms. However, by drawing on a large dataset, which targeted tweets from scientists and researchers, this risk was reduced.

Tweets in our dataset referred to definitions and the differences between modes of research, which indicates that (at least some of) the researchers were familiar with the terminology and different meanings of the three modes of research. In addition, we assume that there are tweets in which the content refers to the three different modes of research, but does not mention the terms explicitly. These tweets were not captured in our dataset. Such tweets are difficult to capture and require a manual analysis and context interpretation, which was not feasible within the bounds of this study.

Another limitation is that 'transdisciplinary' can potentially have two different meanings, i.e. (i) the inclusion of non-academic participants in IDR and (ii) transcending disciplinary boundaries through the development of new methods from two or more scientific disciplines, which could potentially lead to a new discipline. However, based on the manual inspection of the content of tweets, it is assumed that transdisciplinarity is most commonly used with meaning (i) within the dataset.

The use of the seven labelled datasets, containing 71 239 training tweets from various domains, might not appropriately capture the sentiments found within the target tweets. For instance, the word 'difficult' or 'hard' might not necessarily convey a negative sentiment within an academic setting. The inclusion of 1000 manually labelled target tweets during the model training phase captures IDR domain-specific nuances, but might not be able to capture all of them.

The current state of the art in sentiment analysis typically employs neural network models [40]. Specifically, the recurrent neural network model is better designed to handle sequence data, such as text, compared to, for instance, the Multi-layer Perceptron. Recurrent neural networks have the advantage of taking word order into account, and advanced implementations, such as the LSTM model [39], achieve top-ranking results in sentiment classification competitions. Additionally, tweet data are, increasingly, represented as word embeddings [75], a high-dimensional vector representation of words, and have been shown to boost performance of sentiment classification tasks [40,76]. However, such state-of-the-art methods and architectures come with a high degree of complexity, and training these models can be challenging and time consuming. It would be an interesting approach to study in future research, although the purpose of this analysis was not to build top-ranking sentiment classification systems. More importantly, the employed classifier (the SVM) achieves near to state-of-the-art results if properly parametrized [40,76], and would adequately provide a sense of sentiment for the three modes of research studied in this paper.

## 6. Conclusion

For the success of integrative research approaches, it is important to foster positive thinking and optimism through the study of 'bright spots'. Bright spots can help to harvest the full potential of integrative research and enable a feeling of empowerment among researchers engaging in these approaches. This study identified such 'bright spots' by analysing the sentiment of tweets from over 37 000 researchers and scientists on inter-, trans- and multi-disciplinarity, in which they expressed dominantly positive opinions (excluding neutral tweets). Positive opinions were created through, and were based on, positive experiences and successes within integrative research, such as accepted publications, the acquisition of funding, and effective teamwork. As such, this study demonstrates and documents positive thinking within integrative research and gives reason for optimism. The continued study of bright spots and propagation of optimism can potentially have many benefits for integrative research, and hopefully inspire and empower scientists to continue and strive for integrative research in the future.

Data accessibility. The dataset of all extracted tweets are made available in the electronic supplementary material (Data S1). The code that supports this study is openly available on S.S.'s GitHub page at https://github.com/shaheen-syed/Twitter-Sentiment-Analysis.
Authors' contributions. C.T.W. and S.S. retrieved the data. S.S. processed the data and prepared the figures. S.S. analysed the data. C.T.W. and S.S. designed the research, interpreted the results, wrote the manuscript and gave their final approval for publication.
Competing interests. The authors declare no competing interests.

Funding. This research was funded by the project SAF21 – Social Science Aspects of Fisheries for the 21st Century, a project financed under the EU Horizon 2020 Marie Skłodowska-Curie (MSC) ITN-ETN Program; project number: 642080.

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
