## [Reviewer comments · Royal Society Open Science]

Review History

RSOS-190473.R0 (Original submission)

Review form: Reviewer 1

Is the manuscript scientifically sound in its present form?

No

Are the interpretations and conclusions justified by the results?

Yes

Is the language acceptable?

Yes

Is it clear how to access all supporting data?

Yes

Do you have any ethical concerns with this paper?

No

Have you any concerns about statistical analyses in this paper?

No

Recommendation?

Major revision is needed (please make suggestions in comments)

Comments to the Author(s)

Motivated by the much reported negative experiences about interdisciplinary research and the need for the creation of “interdisciplinary optimism,” the manuscript turns to Twitter to search for the expression of positive experiences, as measured by the sentiment of tweets, about inter-/trans-/multi-disciplinary research conveyed there. It used Twitter search API to collect a one-year dataset and trained machine learning classifiers to infer the sentiment of the collected tweets. Further qualitative analysis is presented to describe what the positive/neutral/negative tweets are about. Thorough discussions and limitations are also presented.

I wonder if the following suggestions could be addressed to further improve the paper:

I am not sure about the suitability of applying ML classifiers that are trained using already labeled data from other domains (Table 2) to a target one that is about interdisciplinarity. I would imagine that different domains may exhibit different statistical patterns with which the classifiers try to learn, especially if the domains from which the training data comes are remotely similar to the target domain. Indeed, the presented results seem to suggest that the models may not do a good job in transferring the pattern learned using other domain data to the target tweets---the F1 score (0.83) obtained using the target tweets is much higher than the score using public tweets (0.67; Page 8). I believe that the common pipeline to fulfill the task is to first manually label a sufficiently large sample of the target tweets and then train classifiers and apply them to the remaining unlabeled data, although I understand that in practice, manual labeling is a laborious process.

I also wonder if the data and methods section could be made more concisely to improve the flow of the manuscript (currently it has 6 pages), and if the entire snapshots of tweets are needed (Figs 4-6), or just the text content of the tweets are necessary?

Since the paper has classified users based on their disciplines, it may be interesting to examine the distributions of sentiment by disciplines and to understand whether users from certain disciplines are more likely to post positive/negative tweets.

Minor comments:

- Page 3 Dataset section: Twitter API -> Twitter search API
- Page 4: A random sample of 5,000 tweets -> 5,000 users?

Review form: Reviewer 2**Is the manuscript scientifically sound in its present form?**

Yes

Are the interpretations and conclusions justified by the results?

Yes

Is the language acceptable?

Yes

Is it clear how to access all supporting data?

Yes

Do you have any ethical concerns with this paper?

No

Have you any concerns about statistical analyses in this paper?

No

Recommendation?

Accept with minor revision (please list in comments)

Comments to the Author(s)

Over the past couple years there has been an increasing number of papers that draw on data from social media. When I read them I am quite often sceptical about their methods, analysis, interpretation and robustness. However, this paper - Interdisciplinary optimism? Sentiment analysis of Twitter Data - is the exception (at least from what I've read). This paper has an incredibly rigorous research design, the analysis is perfectly suited to the data, and the interpretation and discussion of the data is very thorough (and includes a section specifically acknowledging the limitations of the approach used, increasing transparency for the readers). The paper is also very well written, and deals with a very important topic (interdisciplinary research) - making this a very timely contribution to the academic literature. I congratulate the authors on doing such a wonderful job, and I only wish that all papers that I was invited to review were this well done!

My recommendation is that this manuscript be accepted for publication following a few very minor revisions:

- Page 8, Section (i) Inspecting tweets: This section would benefit from additional information relating to the 'manual' checking. For example, how was this done, who did it (one author or two authors, or more), etc.
- Page 9, Section (c) Number of User Tweets: The paper states that tweets originated from over 37,000 unique users. I assume the authors meant to say 37,000. If so, this needs updating for accuracy.
- Page 16, sub-section "Positive Tweets": The wording "Positive tweets demonstrated experiences of success stories of the known challenges of interdisciplinary...." was not clear to me. I suggest revising this sentence for clarity.

Decision letter (RSOS-190473.R0)

07-May-2019

Dear Ms Weber,

The editors assigned to your paper ("Interdisciplinary optimism? Sentiment analysis of Twitter Data.") have now received comments from reviewers. We would like you to revise your paper in accordance with the referee and Associate Editor suggestions which can be found below (not including confidential reports to the Editor). Please note this decision does not guarantee eventual acceptance.

Please submit a copy of your revised paper before 30-May-2019. Please note that the revision

deadline will expire at 00.00am on this date. If we do not hear from you within this time then it will be assumed that the paper has been withdrawn. In exceptional circumstances, extensions may be possible if agreed with the Editorial Office in advance. We do not allow multiple rounds of revision so we urge you to make every effort to fully address all of the comments at this stage. If deemed necessary by the Editors, your manuscript will be sent back to one or more of the original reviewers for assessment. If the original reviewers are not available, we may invite new reviewers.

- Data accessibility

<http://datadryad.org/submit?journalID=RSOS&manu=RSOS-190473>

- Competing interests

- Authors' contributions

All submissions, other than those with a single author, must include an Authors' Contributions section which individually lists the specific contribution of each author. The list of Authors should meet all of the following criteria; 1) substantial contributions to conception and design, or

acquisition of data, or analysis and interpretation of data; 2) drafting the article or revising it critically for important intellectual content; and 3) final approval of the version to be published.

- Acknowledgements

- Funding statement

on behalf of Dr Matjaz Perc (Associate Editor) and Marta Kwiatkowska (Subject Editor)
openscience@royalsociety.org

Comments to Author:

Reviewers' Comments to Author:

Reviewer: 1

Comments to the Author(s)

Motivated by the much reported negative experiences about interdisciplinary research and the need for the creation of “interdisciplinary optimism,” the manuscript turns to Twitter to search for the expression of positive experiences, as measured by the sentiment of tweets, about inter-/trans-/multi-disciplinary research conveyed there. It used Twitter search API to collect a one-year dataset and trained machine learning classifiers to infer the sentiment of the collected tweets. Further qualitative analysis is presented to describe what the positive/neutral/negative tweets are about. Thorough discussions and limitations are also presented.

I wonder if the following suggestions could be addressed to further improve the paper:

I am not sure about the suitability of applying ML classifiers that are trained using already labeled data from other domains (Table 2) to a target one that is about interdisciplinarity. I would imagine that different domains may exhibit different statistical patterns with which the classifiers

try to learn, especially if the domains from which the training data comes are remotely similar to the target domain. Indeed, the presented results seem to suggest that the models may not do a good job in transferring the pattern learned using other domain data to the target tweets---the F1 score (0.83) obtained using the target tweets is much higher than the score using public tweets (0.67; Page 8). I believe that the common pipeline to fulfill the task is to first manually label a sufficiently large sample of the target tweets and then train classifiers and apply them to the remaining unlabeled data, although I understand that in practice, manual labeling is a laborious process.

I also wonder if the data and methods section could be made more concisely to improve the flow of the manuscript (currently it has 6 pages), and if the entire snapshots of tweets are needed (Figs 4-6), or just the text content of the tweets are necessary?

Since the paper has classified users based on their disciplines, it may be interesting to examine the distributions of sentiment by disciplines and to understand whether users from certain disciplines are more likely to post positive/negative tweets.

Minor comments:

- Page 3 Dataset section: Twitter API -> Twitter search API
- Page 4: A random sample of 5,000 tweets -> 5,000 users?

Reviewer: 2

Comments to the Author(s)

Over the past couple years there has been an increasing number of papers that draw on data from social media. When I read them I am quite often sceptical about their methods, analysis, interpretation and robustness. However, this paper - Interdisciplinary optimism? Sentiment analysis of Twitter Data - is the exception (at least from what I've read). This paper has an incredibly rigorous research design, the analysis is perfectly suited to the data, and the interpretation and discussion of the data is very thorough (and includes a section specifically acknowledging the limitations of the approach used, increasing transparency for the readers). The paper is also very well written, and deals with a very important topic (interdisciplinary research) - making this a very timely contribution to the academic literature. I congratulate the authors on doing such a wonderful job, and I only wish that all papers that I was invited to review were this well done!

My recommendation is that this manuscript be accepted for publication following a few very minor revisions:

- Page 8, Section (i) Inspecting tweets: This section would benefit from additional information relating to the 'manual' checking. For example, how was this done, who did it (one author or two authors, or more), etc.
- Page 9, Section (c) Number of User Tweets: The paper states that tweets originated from over 37,000 unique users. I assume the authors meant to say 37,000. If so, this needs updating for accuracy.
- Page 16, sub-section "Positive Tweets": The wording "Positive tweets demonstrated experiences of success stories of the known challenges of interdisciplinary...." was not clear to me. I suggest revising this sentence for clarity.

Author's Response to Decision Letter for (RSOS-190473.R0)

See Appendix A.

RSOS-190473.R1 (Revision)

Review form: Reviewer 1

Is the manuscript scientifically sound in its present form?

Yes

Are the interpretations and conclusions justified by the results?

Yes

Is the language acceptable?

Yes

Is it clear how to access all supporting data?

Yes

Do you have any ethical concerns with this paper?

No

Have you any concerns about statistical analyses in this paper?

No

Recommendation?

Accept as is

Comments to the Author(s)

All my comments have been addressed.

Review form: Reviewer 2

Is the manuscript scientifically sound in its present form?

Yes

Are the interpretations and conclusions justified by the results?

Yes

Is the language acceptable?

Yes

Is it clear how to access all supporting data?

Yes

Do you have any ethical concerns with this paper?

No

Have you any concerns about statistical analyses in this paper?

No

Recommendation?

Accept as is

Comments to the Author(s)

Based on the revisions made, I believe that the manuscript is now suitable for publication.

Decision letter (RSOS-190473.R1)

21-Jun-2019

Dear Ms Weber,

I am pleased to inform you that your manuscript entitled "Interdisciplinary optimism? Sentiment analysis of Twitter Data." is now accepted for publication in Royal Society Open Science.

Kind regards,

Lianne Parkhouse

Editorial Coordinator

on behalf of Dr Matjaz Perc (Associate Editor) and Marta Kwiatkowska (Subject Editor)

Reviewer comments to Author:

Reviewer: 2

Based on the revisions made, I believe that the manuscript is now suitable for publication.

Reviewer: 1

All my comments have been addressed.

Appendix A

RSOS Review – Manuscript: “Interdisciplinary Optimism? Sentiment Analysis of Twitter Data.”

Comments from Reviewers and authors’ reply with descriptions of implemented changes.

Comments from Reviewer 1	Author Reply
I am not sure about the suitability of applying ML classifiers that are trained using already labeled data from other domains (Table 2) to a target one that is about interdisciplinarity. I would imagine that different domains may exhibit different statistical patterns with which the classifiers try to learn, especially if the domains from which the training data comes are remotely similar to the target domain. Indeed, the presented results seem to suggest that the models may not do a good job in transferring the pattern learned using other domain data to the target tweets---the F1 score (0.83) obtained using the target tweets is much higher than the score using public tweets (0.67; Page 8). I believe that the common pipeline to fulfill the task is to first manually label a sufficiently large sample of the target tweets and then train classifiers and apply them to the remaining unlabeled data, although I understand that in practice, manual labeling is a laborious process.	Adjustment made to the limitations section to highlight this comment further. Reply: You are indeed right that the best solution would be to manually label a substantial amount of target (int/mult/trans/) tweets, 10,000 – 20,000, and to then train classifiers on this labeled data. Unfortunately, as you also indicated, this is an extremely laborious process, and typically external sources of labeled datasets are used as a common practice. However, by drawing on +70,000 labeled training tweets we are able to capture a wide array of sentiments. The inclusion of 1,000 labeled target tweets allows us to also capture specific sentiment nuances found within the target tweets. This is generally considered a good practice when performing sentiment analysis, but agreed, some limitations will always remain. The reported scores for the target tweets indicate that the classifier is actually doing a better job on the target tweets, which is what we want, since they are the main results of the paper. The reported scores for the test set are there to justify the reason for using an SVM classifier.
I also wonder if the data and methods section could be made more concisely to improve the flow of the manuscript (currently it has 6 pages), and if the entire snapshots of tweets are needed (Figs 4-6), or just the text content of the tweets are necessary?	We have chosen to include some more general text regarding machine learning and sentiment analysis in the methods section so readers not familiar with the methods can follow without having to consult other sources. For instance, 2(e) and 2(f) can be shortened to a short paragraph. Given the multidisciplinary nature of the journal, we believe this is a good thing. We defer to the editor if such explanations are warranted. Regarding the snapshots of the tweet: According to Twitter’s Policies and display requirements, one is required to display any Twitter content according to their guidelines. This means, we cannot simply ‘quote’ the text content because this would violate Twitter’s policy, but have to display the entire tweet as it would appear on

	Twitter. We have decided to include the snapshots of tweets to give readers some insights into ‘real’ examples. This has also been done in a similar way in other papers looking at tweets.
Since the paper has classified users based on their disciplines, it may be interesting to examine the distributions of sentiment by disciplines and to understand whether users from certain disciplines are more likely to post positive/negative tweets.	This would be very interesting indeed ☺ We have performed the analysis and included it in the manuscript within the results (Section (d)) and discussion section under section ‘positive tweets’.
- Page 3 Dataset section: Twitter API -> Twitter search API	Corrected.
- Page 4: A random sample of 5,000 tweets -> 5,000 users?	We replaced the word ‘tweets’ with ‘user descriptions’.
Comments from Reviewer 2	Author Reply
Page 8, Section (i) Inspecting tweets: This section would benefit from additional information relating to the ‘manual’ checking. For example, how was this done, who did it (one author or two authors, or more), etc	We added additional explanations on who did the examination, and explained the examination process in more detail in Section (i).
Page 9, Section (c) Number of User Tweets: The paper states that tweets originated from over 37.000 unique users. I assume the authors meant to say 37,000. If so, this needs updating for accuracy.	Yes, we meant to say 37,000 unique users. This has been corrected.
Page 16, sub-section “Positive Tweets”: The wording “Positive tweets demonstrated experiences of success stories of the known challenges of interdisciplinary....” was not clear to me. I suggest revising this sentence for clarity.	We have revised the sentence for clarity.